# Enablers and Barriers to the Utilization of Antenatal Care Services in India

**DOI:** 10.3390/ijerph16173152

**Published:** 2019-08-29

**Authors:** Felix Akpojene Ogbo, Mansi Vijaybhai Dhami, Ebere Maureen Ude, Praween Senanayake, Uchechukwu L. Osuagwu, Akorede O. Awosemo, Pascal Ogeleka, Blessing Jaka Akombi, Osita Kingsley Ezeh, Kingsley E. Agho

**Affiliations:** 1Translational Health Research Institute (THRI), School of Medicine, Western Sydney University, Campbelltown Campus, Locked Bag 1797, Penrith 2571, Australia; 2General Practice Unit, Prescot Specialist Medical Centre, Welfare Quarters, Makurdi 972261, Nigeria; 3School of Medicine|Diabetes Obesity and Metabolism Translational Research Unit (DOMTRU), Macarthur Clinical School, Parkside Crescent, Campbelltown 2560, Australia; 4Obstetrics and Gynaecology Unit, City Hospital, Emmanuel Mande Close, Makurdi 972261, Nigeria; 5School of Public Health and Community Medicine, Faculty of Medicine, University of New South Wales, Sydney 2052, Australia

**Keywords:** antenatal care, India, factors, Demographic and Health Survey, enablers, barriers

## Abstract

Antenatal care (ANC) reduces adverse health outcomes for both mother and baby during pregnancy and childbirth. The present study investigated the enablers and barriers to ANC service use among Indian women. The study used data on 183,091 women from the 2015–2016 India Demographic and Health Survey. Multivariate multinomial logistic regression models (using generalised linear latent and mixed models (GLLAMM) with the mlogit link and binomial family) that adjusted for clustering and sampling weights were used to investigate the association between the study factors and frequency of ANC service use. More than half (51.7%, 95% confidence interval (95% CI): 51.1–52.2%) of Indian women had four or more ANC visits, 31.7% (95% CI: 31.3–32.2%) had between one and three ANC visits, and 16.6% (95% CI: 16.3–17.0%) had no ANC visit. Higher household wealth status and parental education, belonging to other tribes or castes, a woman’s autonomy to visit the health facility, residence in Southern India, and exposure to the media were enablers of the recommended ANC (≥4) visits. In contrast, lower household wealth, a lack of a woman’s autonomy, and residence in East and Central India were barriers to appropriate ANC service use. Our study suggests that barriers to the recommended ANC service use in India can be amended by socioeconomic and health policy interventions, including improvements in education and social services, as well as community health education on the importance of ANC.

## 1. Introduction

Pregnancy and childbirth complications are the leading causes of maternal mortality worldwide, as an estimated 830 women lose their lives daily from preventable pregnancy- and/or childbirth-related causes. Over 99% of those maternal deaths occur in low- and middle-income countries (LMICs, including India) [1]. Antenatal care (ANC) provides a unique opportunity for screening and diagnosis, health promotion, and disease prevention among pregnant women, and their families and communities [2]. Appropriate utilisation of ANC services corresponds to improved maternal and newborn health, as well as a reduction in maternal deaths during pregnancy and childbirth [3,4,5]. Based on the benefits of ANC, the World Health Organisation (WHO) recommends that pregnant women should attend at least four ANC visits to increase opportunities for risk stratification and/or the identification, prevention, and management of pregnancy and/or comorbidities, as well as health promotion [2].

Worldwide, approximately 64% of women had attended four or more ANC visits in 2016. However, the attainment of the recommended ANC visits varied between and within countries, with LMICs reporting lower percentages [2]. In India, previous reports have indicated that the proportion of women who had four or more ANC visits has increased by approximately 38% over a 10-year period, from 37% in 2006 [6] to 51% in 2016 [7]. While this improvement in ANC service use may be commendable, it also suggests that many Indian women do not achieve the recommended four or more ANC visits, a proxy for comprehensive maternal care during pregnancy [2]. This lack of access to appropriate ANC may have potentially adverse short- and long-term impacts on Indian women and newborns. These adverse effects may include maternal death or health loss from haemorrhage, hypertensive disorders, sepsis, and abortion [4,8], as well as stillbirth and neonatal death [9].

Past nationally representative studies conducted based on the 2005–2006 India Demographic and Health Survey (DHS) data elucidated a number of factors associated with the underutilization of ANC services. These factors included low parental education, urban residence, a lack of mass-media exposure [10], lower household wealth, the region of residence, and belonging to Scheduled Castes, Scheduled Tribes, and the other backward class [11]. In addition, previous subnational studies suggested that financial and cultural issues, as well as a lack of awareness of the benefits of ANC among women and their partners, were also barriers to appropriate ANC service use [12,13]. Nevertheless, it is uncertain whether these factors have changed in the past 10 years in India, given the improvements in household economic and educational status [14], social mobility of women [15,16], and the implementation of the Government of India maternal and child health interventions. These programs included the National Rural Health Mission (2005), the National Urban Health Mission (2008), and the Reproductive, Maternal, Newborn, Child, and Adolescent Health (RMNCH+A) Strategy, introduced in 2013 to improve maternal and child health outcomes, including promotion of ANC service use [17]. Understanding the contextual factors that influence a mother’s decision to attend, or not to attend ANC is crucial to healthcare practitioners and policymakers, as it can offer relevant information and opportunities for targeted policy interventions.

Additionally, findings from the 2005–2006 India DHS may also differ from those obtained from the 2015–2016 DHS, the data source for the present study, due to variations in the sample size and methods used. The 2015–2016 India DHS collected data from 601,509 households, drawn from about 1.2 billion people [7], compared to 110,000 households in the 2005–2006 DHS, drawn from approximately 1 billion people [6]. Also, the 2015–2016 DHS methodology now forms the foundation for future national household surveys in India [7]. The availability of improved methodology for health information gathering and data also suggests the need for up-to-date evidence on the determining factors of ANC service use to guide national maternal health efforts. Therefore, the study aimed to investigate the enablers and barriers to ANC service use in India.

## 2. Methods

### 2.1. Data Sources

The broader methodological approach used in this study has been described in previously published studies [18,19,20]. In summary, the present study was based on the 2015–2016 India DHS, also known as the National Family Health Survey (NFHS-4). The data were collected by the International Institute for Population Sciences (IIPS), supervised by the Ministry of Health and Family Welfare (MoHFW), Government of India. A stratified two-stage sampling design for both rural and urban areas was used to obtain the total sample, based on the 2011 census frame, with villages serving as the primary sampling units (PSU) in rural areas and census enumeration blocks as the PSU in urban areas [7]. Using a standardised questionnaire, maternal and child health information (e.g., antenatal care, delivery care, postnatal care, and infant and young child feeding), as well as sociodemographic characteristics, were collected from eligible women aged 15–49 years in 601,509 households. Respondents were women who were residents in the household 24 h prior to the survey. In these households, 699,686 women were interviewed (204,735 in urban and 494,951 in rural areas), with an overall response rate of 96.7%. Detailed information on the survey methodology is provided in the final India DHS report [7].

In the present study, we used a total weighted sample of the 183,091 most recent live-birth infants of mothers, consistent with the India DHS report [7] and previously published studies [21,22], in an effort to minimise the potential effect of recall bias.

### 2.2. Outcome Variable

The outcome variable was the frequency of ANC service use, based on the WHO recommendation, where ANC service use was measured as at least a visit to the doctor, nurse, midwife, or lay health visitors [7]. In the present study, women were categorised according to their ANC service use into no ANC visits, one to three ANC visits, or four or more ANC visits, and these formed the outcome variables in the analyses. The no ANC visits group formed the reference category of the outcome variables in the analyses.

### 2.3. Study Factors

We adapted the Andersen behavioural conceptual framework [23] to group the study factors potentially related to ANC service use, based on evidence from past studies [18,24,25,26]. Twenty-three study factors were identified and categorised into four main groups: community, predisposing (socio-demographic and health knowledge), enabling, and need factors (Figure 1). Community-level factors included place of residence (urban and rural) and geopolitical region (North, South, East, West, Central, and Northeast), while predisposing factors included health knowledge (frequency of reading magazines or newspapers, frequency of listening to radio, and frequency of watching TV, knowledge of pregnancy complications, knowledge of delivery complications, and knowledge of post-birth complications), and socio-demographic factors (maternal age at delivery, household wealth index, maternal and paternal education, maternal marital status, maternal employment status, maternal body mass index, and types of castes/tribes). Enabling factors included permission to visit health services, distance to health facility, the presence of a companion, getting money to pay for health services, and household decision-making. Need factors included contraceptive use and desire for a pregnancy. In India, the Constitution recognises certain ethnic minority groups for special consideration as Scheduled Tribes, Scheduled Castes, or Other Backward Classes [27,28].

### 2.4. Statistical Analysis

The statistical approach was similar to previous studies [20,26,29]. In this study, preliminary analyses were conducted to describe the survey frequencies, as well as the prevalence of ANC service use and by the study factors in India using the Taylor series linearization to estimate confidence intervals (CIs) around the prevalence estimates. This was followed by the analysis to investigate potential factors related to the use of ANC services in univariate and multivariate multinomial logistic regression models. Generalised linear latent and mixed models (GLLAMM) with the mlogit link and binomial family that adjusted for clustering and sampling weights were used to assess the association between the independent variables and the outcome variables (no ANC visits, one to three ANC visits, or four or more ANC visits). A four-stage model was employed in the multivariate analyses, similar to the adopted conceptual model described by Anderson [23] and used in past studies [18,19]. In the first stage, community-level factors were entered into the model to assess their association with the outcome variables. In the second stage, the significant factors obtained from the stage one model were added to socio-demographic and health service factors to examine their relationship with the outcome variables. The same analytical approach was used for the enabling and need factors in the third and fourth stages, respectively. Any collinearity between the study factors was also investigated but none was evident in the analyses.

For the multivariate model, we calculated and reported statistically significant odds ratios (ORs) and their corresponding 95% CIs as the measure of association between the study factors and ANC service use in India. The analyses were conducted using the ‘svy’ command in Stata version 15.0 (Stata Corp, College Station, TX, USA) to adjust for clustering and sampling weights.

### 2.5. Ethics

The DHS project sought and obtained the required ethical approvals from Ethics Review Board of the International Institute for Population Sciences, Mumbai, India before the surveys were conducted, with informed consent obtained from participants during the surveys. Approval to use the data was sought from Measure DHS and permission was granted.

## 3. Results

### 3.1. Distribution of ANC Service Use by Study Factors

Of the total weighted sample of 183,091 reproductive age maternal responses, 16.6% (95% confidence intervals (95% CI): 16.3–17.0%) of women had no ANC visits, and 31.7% (95% CI: 31.3–32.2%) of women had between one and three ANC visits. A little above half (51.7%, 95% CI: 51.1–52.2%) of Indian women had four or more ANC visits (Table 1). A wide variation in the number of ANC visits was observed across Indian regions, as women who resided in the East reported the highest proportion of non-use of ANC services (39.3%, 95% CI: 38.2–40.6%), while those in the Northeast reported the lowest proportion of non-use of ANC services (2.9%, 95% CI: 2.7–3.2%). Women who resided in the South had the highest percentage of four or more ANC visits (28.4%, 95% CI: 27.7–29.2%), while those in the Northeast reported the lowest (3.7%, 95% CI: 3.5–3.8%) (Table 1).

### 3.2. Factors Associated with ANC Use (1–3 Visits)

Women from Southern, Eastern, Western, and Central India were less likely to make one to three ANC visits compared to those who resided in Northern India. Women from wealthier households were more likely to have between one and three ANC visits compared to those from poorer households. Women with higher education were more likely to make 1–3 ANC visits compared to those with no education (adjusted odds ratio column, Table 2). Exposure to the media (newspapers or magazines and television) was associated with Indian women attending between one and three ANC visits compared to non-media exposure. Women who reported difficulties in seeking permission to visit health services were less likely to attend one to three ANC visits compared to those who did not have problems in seeking permission (adjusted odds ratio column, Table 2).

### 3.3. Factors Associated with ANC Use (≥4 Visits)

Women from Southern India were more likely to attend four or more ANC visits compared to those from Northern India. In contrast, women from Eastern and Central India were less likely to attend four or more ANC visits compared to those from Northern India (adjusted odds ratio column, Table 2). Women from wealthier households were more likely to receive the four or more ANC visits compared to those from poorer households. In addition, higher parental education was associated with the attainment of four or more ANC visits compared to no maternal or partner education. Women from other tribes or castes were more likely to attend four or more ANC visits compared to those from Scheduled Caste. Women who engaged with the media (newspapers or magazines and television) were more likely to attend four or more ANC visits compared to those who did not engage with the media. Women who reported having problems seeking permission to attend health services and those who were not involved in household decision-making were less likely to attend four or more ANC visits compared to their counterparts. Women who reported visiting health facilities unaccompanied were less likely to attend four or more ANC visits compared to those who were accompanied by someone (adjusted odds ratio column, Table 2).

## 4. Discussion

The present study showed that 16.6% of Indian women had no ANC visits, 31.7% had between one and three ANC visits, and just over half received the recommended four or more ANC visits (51.7%). The study demonstrated that higher household wealth status and parental education, belonging to other tribes or castes, a woman’s autonomy to visit the health facility, residence in Southern India, as well as exposure to the media were enablers of the recommended ANC service use (≥4 visits). The recommended four or more ANC attendance was also associated with contraceptive use and a woman’s desire for pregnancy. In contrast, lower household wealth, a lack of woman’s autonomy, and residence in Eastern and Central India were barriers to appropriate ANC use.

The study indicated that mothers who resided in Southern India were more likely to attend at least four ANC visits, as recommended by the WHO compared to their counterparts in Northern India [2]. In contrast, mothers who resided in the East, Northeast, and Central India were less likely to attend at least four ANC visits compared to those who reside in Northern India. To the authors’ knowledge, region-specific determinants of ANC service use have not been elucidated in India; however, possible reasons for the regional variations in ANC use may be due to region-specific differences in economy, education, access and distance to health facilities, as well as the quality of service provision [30,31]. Studies that assess region-specific determining factors of ANC uptake may be needed to inform a more equitable distribution of maternal and child health (MCH) resources and policies at the sub-national level in India.

The association between household wealth status and ANC service utilisation has been documented in LMICs [32,33], and was a key factor in the present study. We found that mothers from wealthier households were more likely to attend ANC services compared to those from poorer households, and this association was stronger in those who attended at least four ANC visits. These findings are consistent with previous studies conducted in India [34,35], as well as other LMICs, including Nigeria [18], Pakistan [36], and Kenya [37]. Mothers who belonged to lower wealth quintiles may have greater financial challenges in accessing ANC services, as reported in studies from regional India [12,13]. Although ANC is almost free in most public hospitals in India, issues such as poor infrastructure, health professional absenteeism, and a shortage of medications in public hospitals, particularly in rural areas, may have pushed pregnant women to seek private ANC services [38]. Similarly, the costs associated with attending private maternity facilities [39] and the cost of transportation [40] have also been described as key barriers to accessible ANC in India.

In recent years, India has implemented a number of MCH interventions to tackle the expenses associated with maternity care services. Most notable of these is the Janani Suraksha Yojana (JSY) scheme, which was implemented by the National Rural Health Mission (NRHM) in 2005 to provide underprivileged pregnant women with cash assistance [41]. Nevertheless, the costs associated with accessing ANC remains significantly higher than the subsidies provided by programs such as the JSY program [42]. To improve ANC uptake among Indian women and other MCH outcomes, the Government of India has recently launched new MCH schemes, including the Pradhan Mantri Matru Vandana Yojana, Pradhan Mantri Surakshit Matritva Abhiyan, and LaQshya programmes [43,44]. While these initiatives are useful and required, the assessment of how successful and impactful these programs are may need to be documented in the scientific literature, consistent with a previous program [17], to guide future MCH programmes in India.

Higher maternal education level is an important enabler of ANC utilisation in the present study, as mothers with secondary or higher schooling were more likely to attend at least four ANC visits. The dose-response relationship between maternal education and ANC service use observed in this study is consistent with previous studies conducted in India [10], Indonesia [45], Bangladesh [46], and Turkey [47]. Higher maternal education may have a synergistic effect with other enablers of ANC utilisation, as women with higher education may be more likely to live in urban areas, gain employment, possess more wealth, and have a better understanding of the benefits of attending ANC [24]. This association was mirrored in the partner education levels, potentially due to similar reasons. Furthermore, higher education may empower parents to make informed decisions about their health and take action on health promotion initiatives. Our study highlights the importance of targeting low education mothers with health promotion messages, as the majority of mothers who did not attend ANC had no education. More broadly, the Government of India may need to ensure that young girls and boys have access to inclusive and quality education, and ensure a higher completion rates, as articulated in Sustainable Development Goal–4 [48], which may subsequently lead to greater utilisation of ANC service in the long-term.

Consistent with previous reports [18,49], the present study indicated that women who considered access to enabling factors (e.g., household decision-making power, autonomy to attend ANC, and requiring a companion to attend ANC) a big problem, had a corresponding underutilization of ANC services in India. In particular, women who were not involved in household decision-making were less likely to attend between one and three ANC visits, and even less likely to attend four or more ANC visits compared to their counterparts. Similarly, women who reported needing to seek permission from their partners to attend ANC and those who were not usually accompanied to health facilities were also less likely to attend the recommended number of ANC visits. These findings suggest that a woman’s autonomy and support from their partner play important roles in ANC service use.

Contrary to past studies [37,50,51], distance to health facilities was not associated with ANC service use in our study. The Government of India health initiative (the National Rural Health Mission) [17] that expanded MCH services to disadvantaged rural areas may have played a role in our finding. Women’s exposure to mass media (newspapers or magazines and television) was associated with ANC service use, and this is consistent with past studies [10,18]. A lack of exposure to these media outlets may have resulted in women missing out on health promotion messages relating to the benefits of ANC. Additionally, the ownership of media devices may be a direct result of higher household wealth, which was also related to ANC service use. Our research underpins the need to improve women’s autonomy in the household, as well as increase the reach and impact of health promotion campaigns and access to media sources for vulnerable women.

Past studies have suggested that the use of ANC services is influenced by a woman’s desire for pregnancy, as women who carry unplanned pregnancies were less likely to attend ANC [37,52]. This finding was demonstrated in our study, where women who had no desire for pregnancy were less likely to attend the recommended four or more ANC visits compared to those who had no desire for pregnancy. Similarly, women who did not use contraceptives were less likely to attend four or more ANC visits. These findings indicate the need for a scale-up of accessible family planning to women of reproductive age, as well as greater access to contraceptive methods and education in India.

### Study Limitations and Strengths

This study had limitations. First, the study was based on cross-sectional data, which makes an assessment of a clear temporal relationship between the study factors and ANC attendance impossible. Second, the ANC data collected during the NFHS-4 would have been subjected to recall bias, as it relied on self-reporting. This may have resulted in misclassification measurement bias, and subsequently, led to either an over- or under-estimation of the effect size between the study factors and ANC service use. Third, there was a lack of assessment of other potential confounders (e.g., data on health care access or health status of pregnant women), which may have provided additional information with the enablers and barriers to ANC service use in India. The study also had strengths. First, the large representative sample, with a high response rate (approximately 98%). This implies that selection bias may be unlikely to affect the observed results. Second, trained personnel with validated questionnaires were used to collect data in the NFHS-4, which would have strengthened the internal validity of the study. Lastly, the study provides insight into key determinants of ANC visits in India, and thus, provides an opportunity for policymakers and public health practitioners to design and implement focused MCH interventions.

## 5. Conclusions

The present study indicated that higher household wealth status and parental education, belonging to other tribes or castes, a woman’s autonomy to visit the health facility, residence in Southern India, and desire for pregnancy, as well as exposure to the media, were enablers to frequent ANC service use. In contrast, residence in the East, Northeast, and Central regions of India were barriers to the utilisation of ANC services. Also, women from poorer households, those with no or primary education, and women who were not exposed to mass media were less likely to use ANC services compared to their counterparts. Our study elucidates key enablers and barriers to ANC attendance in India. It is vital that current and/or future MCH initiatives focus on women with socioeconomic vulnerabilities, while also designing and implementing multi-pronged MCH interventions that aimed to increase ANC uptake among underserved Indian women.

## Figures and Tables

**Figure 1 ijerph-16-03152-f001:**
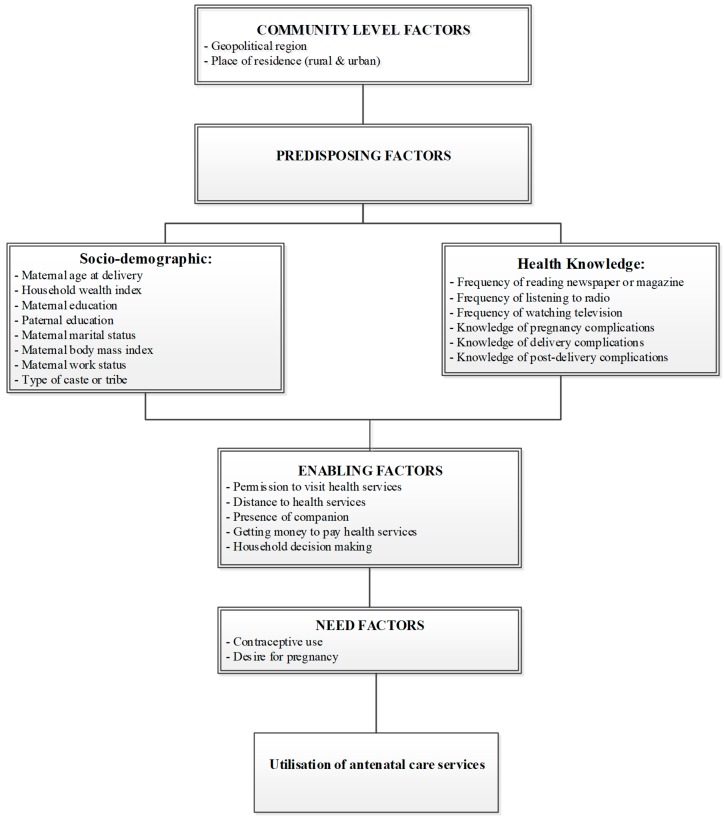
Conceptual framework modified from the Andersen behavioural model [23].

**Table 1 ijerph-16-03152-t001:** Prevalence (PR) with corresponding 95% confidence interval (95% CI) of the frequency of antenatal care service use in India, 2015–2016 NFHS–4 (National Family Health Survey) (*N* = 183,091).

Antenatal Care (ANC) Visits	No ANC Visits		ANC (1–3) Visits		ANC (≥4) Visits	
Variable	*n* *	PR (95% CI)	*n* *	PR (95% CI)	*n* *	PR (95% CI)
Outcomes	30,449	16.6 (16.3–17.0)	58,082	31.7 (31.3–32.2)	94,560	51.7 (51.1–52.2)
Community-level factors						
Residence type						
Urban	5074	16.6 (15.5–17.8)	12,799	22.0 (21.0–23.1)	36,410	38.5 (37.3–39.6)
Rural	25,375	83.3 (82.1–84.4)	45,283	77.9 (76.9–78.9)	58,150	61.5 (60.3–62.6)
Geopolitical region						
North	2705	8.9 (8.2–9.5)	8254	14.2 (13.7–14.7)	11,765	12.4 (12.0–12.9)
South	1977	6.5 (5.7–7.3)	5255	9.0 (8.4–9.7)	26,886	28.4 (27.7–29.1)
East	11,994	39.3 (38.1–40.6)	15,102	26.0 (25.3–26.6)	19,518	20.6 (19.9–21.3)
West	2337	7.7 (6.9–8.4)	4334	7.4 (6.7–8.2)	17,390	18.3 (17.6–19.1)
Central	10,532	34.6 (33.4–35.7)	22,475	38.7 (37.9–39.4)	15,540	16.4 (15.9–16.8)
Northeast	904	2.9 (2.7–3.2)	2661	4.5 (4.3–4.8)	3461	3.6 (3.5–3.8)
Socio-demographic factors						
Household wealth index						
Poor	22,029	72.3 (71.3–73.3)	31,517	54.2 (53.4–55.0)	28,107	29.7 (29.0–30.4)
Middle	4136	13.5 (13.0–14.1)	11,295	19.4 (18.9–19.9)	21,004	22.2 (21.7–22.7)
Rich	4284	14.0 (13.2–14.9)	15,270	26.2 (25.5–27.0)	45,449	48.0 (47.2–48.8)
Mother’s education						
No education	16,481	54.1 (53.1–55.0)	19,914	34.2 (33.6–34.9)	14,276	15.1 (14.6–15.5)
Primary	4299	14.1 (13.6–14.6)	9081	15.6 (15.2–16.0)	11,271	11.9 (11.5–12.2)
Secondary or higher	9669	31.7 (30.8–32.7)	29,087	50.0 (49.3–50.7)	69,014	72.9 (72.4–73.5)
Mother’s working status						
Did not work	3971	13.0 (12.3–13.7)	7920	13.6 (13.1–14.1)	14,756	15.6 (15.0–16.1)
Worked	913	2.9 (2.7–3.2)	1569	2.7 (2.5–2.9)	2728	2.8 (2.6–3.0)
Mother’s age						
15–19 years	927	3.0 (2.7–3.3)	1910	3.2 (3.1–3.4)	3402	3.5 (3.3–3.8)
20–34 years	24,677	81.0 (80.4–81.6)	50,646	87.2 (86.8–87.5)	84,389	89.2 (88.9–89.5)
35–49 years	4845	15.9 (15.3–16.4)	5526	9.5 (9.2–9.8)	6769	7.1 (6.8–7.4)
Maternal BMI kg/m^2^						
≤18	6616	21.7 (21.1–22.3)	11,793	20.3 (19.8–20.7)	14,948	15.8 (15.4–16.2)
19–24	20,543	67.4 (66.7–68.1)	38,719	66.6 (66.1–67.1)	58,416	61.7 (61.2–62.3)
≥25	2745	9.0 (8.5–9.5)	6758	11.6 (11.2–12.0)	19,093	20.1 (19.7–20.6)
Type of caste or tribe						
Scheduled caste	6988	22.9 (22.1–23.8)	12,820	22.0 (21.4–22.6)	19,030	20.1 (19.5–20.7)
Scheduled tribe	3750	12.3 (11.6–12.9)	6342	10.9 (10.4–11.4)	8669	9.1 (8.7–9.5)
Other backward class	14,243	46.7 (45.7–47.8)	26,788	46.1 (45.3–46.9)	38,831	41.0 (40.3–41.7)
Others ^#^	5468	17.9 (17.1–18.8)	12,132	20.8 (20.2–21.6)	28,030	29.6 (28.9–30.3)
Marital status						
Currently married	29,926	98.2 (98.0–98.4)	57,365	98.7 (98.6–98.8)	93,289	98.6 (98.5–98.7)
Formerly married (divorced/separated/widowed)	465	1.5 (1.3–1.7)	670	1.1 (1.0–1.2)	1194	1.2 (1.1–1.3)
Partner education						
No education	1614	5.3 (4.9–5.7)	1852	3.1 (2.9–3.4)	1776	1.8 (1.7–2.0)
Primary	2950	9.6 (9.1–10.2)	6515	11.2 (10.7–11.6)	12,192	12.8 (12.4–13.3)
Secondary or higher	299	0.9 (0.8–1.1)	1089	1.8 (1.7–2.0)	3471	3.6 (3.4–3.9)
Religion						
Hindu	23,825	78.2 (77.1–79.2)	46,598	80.2 (79.4–80.9)	74,029	78.2 (77.5–79.0)
Muslim	5679	18.6 (17.6–19.6)	9233	15.9 (15.1–16.6)	14,511	15.3 (14.6–16.0)
Christian and others	945	3.1 (2.7–3.5)	2250	3.8 (3.6–4.1)	6020	6.3 (5.9–6.7)
Health knowledge factors						
Reads newspaper or magazine						
Not all	25,657	84.2 (83.5–84.9)	41,835	72.0 (71.2–72.7)	51,812	54.7 (54.1–55.4)
Yes	4792	15.7 (15.0–16.4)	16,246	27.9 (27.2–28.7)	42,749	45.2 (44.5–45.8)
Listens to radio						
Not all	26,365	86.5 (85.9–87.2)	50,716	87.3 (86.7–87.8)	80,818	85.4 (85.0–85.9)
Yes	4084	13.4 (12.7–14.0)	7366	12.6 (12.1–13.2)	13,742	14.5 (14.0–14.9)
Watches television						
Not all	17,496	57.4 (56.4–58.4)	21,168	36.4 (35.7–37.1)	13,841	14.6 (14.2–15.0)
Yes	12,953	42.5 (41.5–43.5)	36,914	63.5 (62.8–64.2)	80,719	85.3 (84.9–85.7)
Told about delivery complications						
Any complications	0	0	30,726	52.9 (52.1–53.6)	66,918	70.7 (70.1–71.3)
None	30,449	100	27,356	47.1 (46.3–47.8)	27,642	29.2 (28.6–29.8)
Knowledge of post-delivery complications						
Yes	4911	16.1 (15.4–16.8)	20,235	34.8 (34.1–35.5)	42,506	44.9 (44.2–45.6)
None	25,537	83.8 (83.1–84.5)	37,847	65.1 (64.4–65.8)	52,054	55.0 (54.3–55.7)
Enabling factors						
Household decision making						
Mother involved	3621	11.8 (11.2–12.5)	7566	13.0 (12.5–13.5)	14,721	15.5 (15.0–16.1)
Mother not involved	26,828	88.1 (87.4–88.7)	50,516	86.9 (86.4–87.4)	79,839	84.4 (83.8–84.9)
Seek permission to visit health services						
No problem	13,533	44.4 (43.4–45.4)	31,658	54.5 (53.7–55.2)	58,785	62.1 (61.4–62.8)
Big problem	9207	30.2 (29.3–31.2)	12,447	21.4 (20.8–22.0)	14,468	15.3 (14.8–15.8)
Not a big problem	7709	25.3 (24.5–26.1)	13,977	24.0 (23.4–24.7)	21,307	22.5 (21.9–23.1)
Getting money to pay health services						
No problem	9186	30.1 (29.2–31.0)	23,053	39.6 (38.9–40.4)	45,309	47.9 (47.1–48.6)
Big problem	11,842	38.8 (37.9–39.8)	16,742	28.8 (28.1–29.4)	21,002	22.2 (21.6–22.8)
Not a big problem	9421	30.9 (30.1–31.7)	18,287	31.4 (30.8–32.1)	28,249	29.8 (29.2–30.4)
Distance to health facility						
No problem	6485	21.3 (20.4–22.1)	16,546	28.4 (27.8–29.1)	38,332	40.5 (39.8–41.2)
Big problem	13,823	45.4 (44.3–46.4)	20,906	35.9 (35.2–36.7)	24,232	25.6 (25.0–26.2)
Not a big problem	10,141	33.3 (32.4–34.1)	20,629	35.5 (34.9–36.1)	31,996	33.8 (33.2–34.4)
Accompany to health facility						
No problem	9796	32.1 (31.2–33.1)	23,339	40.1 (39.5–40.8)	49,195	52.0 (51.3–52.7)
Big problem	10,163	33.3 (32.4–34.3)	14,285	24.5 (23.9–25.2)	15,511	16.4 (15.9–16.8)
Not a big problem	10,490	34.4 (33.6–35.3)	20,458	35.2 (34.5–35.8)	29,854	31.5 (30.9–32.1)
Need factors						
Contraceptive use						
Yes	8264	27.1 (26.3–27.9)	23,147	39.8 (39.2–40.4)	46,599	49.2 (48.6–49.8)
No	22,179	72.8 (72.0–73.6)	34,926	60.1 (59.5–60.7)	47,942	50.7 (50.1–51.3)
Wanted pregnancy at the time						
Then	25,844	84.8 (84.2–85.5)	52,189	89.8 (89.5–90.2)	88,139	93.2 (92.9–93.4)
Later	1280	4.2 (3.8–4.5)	2631	4.5 (4.3–4.7)	3582	3.7 (3.5–3.9)
No more	3223	10.5 (10.0–11.1)	3261	5.6 (5.3–5.8)	2839	3.0 (2.8–3.1)

*n* * = the weighted total number varies between categories due to missing data; PR = prevalence of pregnant women who attended ANC across the study variables; CI: confidence interval; others ^#^ includes Jews, Parsis/Zoroastrians, those following “other” religions, and those with no religion.

**Table 2 ijerph-16-03152-t002:** Factors associated with antenatal care service use in India, 2015–2016 NFHS (*N* = 183,091).

Study Factors	Antenatal Care (1–3) Visits						Antenatal Care (≥4) Visits					
	COR	(95% CI)	*p* Value	aOR	(95% CI)	*p* Value	COR	(95% CI)	*p* Value	aOR	(95% CI)	*p* Value
Community-level factors												
Geopolitical region												
North	1.00			1.00			1.00			1.00		
South	0.80	0.74–1.02	0.097	0.72	0.54–0.97	0.035	3.12	2.68–3.65	*p* < 0.001	2.41	1.83–3.16	*p* < 0.001
East	0.41	0.37–0.45	*p* < 0.001	0.47	0.39–0.56	*p* < 0.001	0.37	0.33–0.41	*p* < 0.001	0.40	0.33–0.48	*p* < 0.001
West	0.60	0.51–0.71	*p* < 0.001	0.49	0.37–0.65	*p* < 0.001	1.71	1.49–1.95	*p* < 0.001	1.13	0.87–1.47	0.324
Central	0.69	0.63–0.76	*p* < 0.001	0.76	0.64–0.91	0.003	0.34	0.30–0.37	*p* < 0.001	0.36	0.30–0.44	*p* < 0.001
North East	0.96	0.85–1.09	0.568	1.08	0.84–1.39	0.506	0.88	0.77–1.00	0.057	0.78	0.60–1.01	0.068
Socio-demographic factors												
Household wealth index												
Poor	1.00			1.00			1.00			1.00		
Middle	1.90	1.80–2.02	*p* < 0.001	1.28	1.11–1.48	0.001	3.98	3.74–4.22	*p* < 0.001	1.86	1.61–2.15	*p* < 0.001
Rich	2.49	2.30–2.68	P < 0.001	1.56	1.29–1.88	*p* < 0.001	8.31	7.71–8.96	*p* < 0.001	2.99	2.47–3.62	*p* < 0.001
Mother education												
No education	1.00			1.00			1.00			1.00		
Primary	1.74	1.65–1.84	*p* < 0.001	1.48	1.29–1.70	*p* < 0.001	3.02	2.85–3.21	*p* < 0.001	2.03	1.75–2.36	*p* < 0.001
Secondary and higher	2.49	2.36–2.61	*p* < 0.001	1.77	1.57–2.00	*p* < 0.001	8.24	7.81–8.69	*p* < 0.001	3.42	3.01–3.87	*p* < 0.001
Type of caste or tribe												
Scheduled caste	1.00			1.00			1.00			1.00		
Scheduled tribe	0.92	0.85–0.99	0.039	1.16	0.98–1.38	0.071	0.84	0.78–0.92	*p* < 0.001	1.28	1.05–1.54	0.110
Other backward class	1.02	0.96–1.08	0.381	1.03	0.91–1.17	0.556	1.00	0.94–1.06	0.970	0.88	0.77–1.00	0.058
Others	1.20	1.12–1.30	*p* < 0.001	1.18	0.99–1.40	0.055	1.88	1.74–2.03	*p* < 0.001	1.60	1.34–1.91	*p* < 0.001
Partner education												
No education	1.00			1.00			1.00			1.00		
Primary	1.92	1.73–2.14	*p* < 0.001	1.32	1.19–1.48	*p* < 0.001	3.75	3.35–4.20	*p* < 0.001	1.55	1.37–1.75	*p* < 0.001
Secondary and higher	3.17	2.55–3.95	*p* < 0.001	1.62	1.28–2.04	*p* < 0.001	10.55	8.47–13.13	*p* < 0.001	2.51	1.98–3.18	*p* < 0.001
Health knowledge factors												
Reads newspaper or magazine												
Not all	1.00			1.00			1.00			1.00		
Yes	2.07	1.95–2.21	*p* < 0.001	1.18	1.00–1.38	0.042	4.41	4.17–4.67	*p* < 0.001	1.27	1.08–1.50	0.003
Listens to radio												
Not all	1.00			1.00			1.00			1.00		
Yes	0.93	0.87–1.00	0.080	0.67	0.57–0.78	*p* < 0.001	1.09	1.02–1.17	0.005	0.65	0.55–0.77	*p* < 0.001
Watches television												
Not all	1.00			1.00			1.00			1.00		
Yes	2.35	2.24–2.46	*p* < 0.001	1.42	1.27–1.59	*p* < 0.001	7.87	7.47–8.30	*p* < 0.001	2.56	2.26–2.89	*p* < 0.001
Enabling factors												
Household decision making												
Mother involved	1.00			1.00			1.00			1.00		
Mother not involved	0.90	0.84–0.96	0.002	0.84	0.74–0.95	0.007	0.73	0.68–0.78	*p* < 0.001	0.68	0.59–0.78	*p* < 0.001
Seeking permission to visit health services												
No problem	1.00			1.00			1.00			1.00		
Big problem	0.57	0.54–0.61	*p* < 0.001	0.74	0.64–0.86	*p* < 0.001	3.62	0.34–0.38	*p* < 0.001	0.53	0.45–0.63	*p* < 0.001
Not a big problem	0.77	0.73–0.82	*p* < 0.001	0.79	0.69–0.90	0.001	0.63	0.60–0.67	*p* < 0.001	0.65	0.56–0.76	*p* < 0.001
Getting money to pay health services												
No problem	1.00			1.00			1.00			1.00		
Big problem	0.56	0.53–0.59	*p* < 0.001	0.99	0.84–1.17	0.996	0.36	0.33–0.38	*p* < 0.001	1.08	0.89–1.29	0.406
Not a big problem	0.77	0.73–0.81	*p* < 0.001	0.91	0.79–1.05	0.223	0.60	0.57–0.64	*p* < 0.001	0.90	0.78–1.05	0.215
Distance to health facility												
No problem	1.00			1.00			1.00			1.00		
Big problem	0.59	0.55–0.63	*p* < 0.001	0.99	0.85–1.16	0.934	0.29	0.27–0.31	*p* < 0.001	0.85	0.71–1.01	0.068
Not a big problem	0.79	0.75–0.84	*p* < 0.001	0.97	0.83–1.14	0.779	0.53	0.50–0.56	*p* < 0.001	0.91	0.76–1.08	0.284
Accompanied to health facility												
No problem	1.00			1.00			1.00			1.00		
Big problem	0.59	0.55–0.62	*p* < 0.001	0.90	0.77–1.06	0.222	0.30	0.28–0.32	*p* < 0.001	0.78	0.65–0.94	0.011
Not a big problem	0.81	0.77–0.86	*p* < 0.001	1.06	0.92–1.21	0.385	0.56	0.53–0.59	*p* < 0.001	0.92	0.79–1.06	0.279
Need factors												
Contraceptive use												
Yes	1.00			1.00			1.00			1.00		
No	0.56	0.53–0.58	*p* < 0.001	0.70	0.63–0.78	*p* < 0.001	0.38	0.36–0.40	*p* < 0.001	0.51	0.45–0.57	*p* < 0.001
Wanted pregnancy at the time												
Then	1.00			1.00			1.00			1.00		
Later	1.01	0.92–1.12	0.712	1.02	0.80–1.31	0.822	0.82	0.74–0.90	*p* < 0.001	0.94	0.73–1.21	0.655
No more	0.50	0.46–0.53	*p* < 0.001	0.72	0.61–0.87	0.001	0.25	0.23–0.28	*p* < 0.001	0.51	0.41–0.64	*p* < 0.001

Statistically significant (using *p* value < 0.05 and confidence intervals) study factors from multivariable models are shown. COR: crude odds ratio; aOR: adjusted odds ratio. In the model of community-level factors, adjustments were conducted for predisposing (sociodemographic and health), enabling, and need factors. Similar approaches were used for the predisposing, enabling, and need factors, with adjustments for respective factors in multivariable models.

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
