# Peer review of "Enablers and Barriers to the Utilization of Antenatal Care Services in India"

_ijerph, 2019, doi:10.3390/ijerph16173152_

Round 1

Reviewer 1 Report

I find this manuscript relevant, organized, and clear. Notwithstanding, I believe that several changes should be considered regarding data analysis. I leave a few suggestions that, in my view, could further improve the quality of the manuscript.

Introduction

First paragraph: Statistical data should be provided regarding maternal mortality across the world, particularly in India. Second paragraph: Statistical data should be provided regarding the maternal and newborn complications associated with lack of access to ANC in India.

Methods

Page 3, first paragraph: the authors should clarify whether being a woman aged between 15 and 49 years was the only inclusion criterion for the study. Also, it should be stated how many of the approached women had had a child recently enough to participate in the study. Page 3, second paragraph: the authors should clarify how much time has passed since the birth that the participants considered when answering the questions. Page 3, fourth paragraph: the authors should briefly explain each component of the modified Andersen behavioural conceptual framework. In addition, the authors should explain how the factors potentially related to ANC use were selected. Page 3, Figure 1: the authors should clarify whether the variables “frequency of reading newspaper or magazine/listening to radio/watching television” were specifically related to health information.

Results

First paragraph: there are slight differences regarding the statistical data presented in the text and those presented in Table 1. Table 1: the percentages are not correct, as they should be organized by factor, not by outcome. The current data may reflect differences in the number of people in each category of each factor. For instance, as more than two thirds of the sample lived in rural areas, the relevant question is not “from those who had no ANC visits, how many were from rural areas and how many were from urban areas?”, but instead “considering those living in rural areas, how many had no ANC visits, how many had 1-3 visits and how many had 4 or more visits?” (the same question applies to those living in urban areas). Page 6: the text is easier to follow if the authors comment the data by the same order they are presented in Table 2. Table 2: I believe the odds presented concern “0 antenatal visits” as the reference category of the outcome variable. That should be explicitly stated. Also, I believe that it could be more informative to consider “4 or more antenatal visits” as the reference category. The pattern of results when comparing “0 antenatal visits” to “1-3 antenatal visits” and “4 or more antenatal visits” is largely similar. I would expect more differences if “4 or more antenatal visits” was compared with “0 antenatal visits” and with “1-3 antenatal visits”. Pages 6-14: the authors should specify whether their focus is on crude odds ratios or adjusted odds ratios.

Discussion

Page 11: The term “internal validity” is not correctly used, as it is only present in experimental studies. The authors are referring to construct validity.

Author Response

I find this manuscript relevant, organized, and clear. Notwithstanding, I believe that several changes should be considered regarding data analysis. I leave a few suggestions that, in my view, could further improve the quality of the manuscript.

Response:

We thank the reviewer for the comment, and relevant suggestions are addressed below.

Introduction

First paragraph: Statistical data should be provided regarding maternal mortality across the world, particularly in India.

Response:

Points appreciated and now reflected in the revised manuscript (Page 1, Paragraph 1)

Pregnancy and childbirth complications are the leading causes of maternal mortality worldwide, as approximately 830 women lose their lives daily from preventable pregnancy- and/or childbirth-related causes. Over 99% of those maternal deaths occur in low- and middle-income countries (LMICs, including India) [1].

Second paragraph: Statistical data should be provided regarding the maternal and newborn complications associated with lack of access to ANC in India.

Response:

Points appreciated and now reflected in the revised manuscript (Page 2, Paragraph 1):

This lack of access to appropriate ANC may have potential adverse short- and long-term impacts on Indian women and newborns. These adverse effects may include maternal death or health loss from haemorrhage, hypertensive disorders, sepsis and abortion [4, 8], as well as stillbirth and neonatal death [9].

Methods

Page 3, first paragraph: the authors should clarify whether being a woman aged between 15 and 49 years was the only inclusion criterion for the study. Also, it should be stated how many of the approached women had had a child recently enough to participate in the study.

Response:

The DHS focused on women aged 15–49 years who were resident in the household 24-hour prior to the survey. The number of maternal responses (N=183,091) and information relating to inclusion criteria were noted in original manuscript (Page 3, Paragraph 1):

Respondents were women who were residents in the household 24 hours prior to the survey. In these households, 699,686 women were interviewed (204,735 in urban and 494,951 in rural areas), with an overall response rate of 96.7%. Detailed information on the survey methodology is provided in the final India DHS report [7].

In the present study, we used a total weighted sample of 183,091 most recent live-birth infants of mothers, consistent with the India DHS report [7] and previously published studies [21, 22], in an effort to minimise the potential effect of recall bias.

Page 3, second paragraph: the authors should clarify how much time has passed since the birth that the participants considered when answering the questions.

Response:

In the present study, we focused on the most recent live-birth infants of mothers, consistent with the India DHS report [6] and previously published studies [18, 19]. Thus, it is approximately one year since the birth that repondents answered the questions. Please note that this is a better estimate given that the surveys consider mothers who have children younger than five years as “participants”.

Page 3, fourth paragraph: the authors should briefly explain each component of the modified Andersen behavioural conceptual framework.

Response:

Point appreciated and now reflected in the revised manuscript (Page 3, Paragraph 4):

Community-level factors included place of residence (urban and rural) and geopolitical region (North, South, East, West, Central and North East), while predisposing factors included health knowledge (frequency of reading magazine or newspaper, frequency of listening radio and frequency of watching TV, knowledge of pregnancy complications, knowledge of delivery and knowledge of post-birth complications) and socio-demographic factors (maternal age at delivery, household wealth index, maternal and paternal education, maternal marital status, maternal employment status, maternal body mass index and types of castes/tribes). Enabling factors included permission to visit health services, distance to health facility, the presence of a companion, getting money to pay for health services and household decision-making. Need factors included contraceptive use and desire for a pregnancy. In India, the Constitution recognises certain ethnic minority groups for special consideration as Scheduled Tribes, Scheduled Castes or other backward classes [24, 25].

In addition, the authors should explain how the factors potentially related to ANC use were selected.

Response

This was an omission on our part. Additional text has been incorporated into the revised manuscript in response to the reviewer comment (Page 3, Paragraph 4):

We adapted the Andersen behavioural conceptual framework [20] to group the study factors potentially related to ANC service use based on evidence from past studies [15, 21-23].

Page 3, Figure 1: the authors should clarify whether the variables “frequency of reading newspaper or magazine/listening to radio/watching television” were specifically related to health information.

Response

The frequency of reading newspaper or magazine/listening to radio/watching television was not specific to health information as noted in the NFHS-4 final report.

Reference: International Institute for Population Sciences (IIPS), ICF. National Family Health Survey (NFHS-4), India. Mumbai, India IIPS, 2017.

Results

First paragraph: there are slight differences regarding the statistical data presented in the text and those presented in Table 1. Table 1: the percentages are not correct, as they should be organized by factor, not by outcome. The current data may reflect differences in the number of people in each category of each factor. For instance, as more than two thirds of the sample lived in rural areas, the relevant question is not “from those who had no ANC visits, how many were from rural areas and how many were from urban areas?”, but instead “considering those living in rural areas, how many had no ANC visits, how many had 1-3 visits and how many had 4 or more visits?” (the same question applies to those living in urban areas).

Response

We note that the data presented are correct as they relate to the study question. We agree with the reviewer that an additional question could be asked which may relate to “considering those living in rural areas, how many had no ANC visits, how many had 1-3 visits and how many had 4 or more visits?” However, our aim was to consider the enablers and barrier to ANC use in India, with no specific reference to rural-urban differences. We note that a manuscript that considered national and rural-urban determinants of early initiation of breastfeeding in India (Senanayake et al., 2016) has recently been published, in reponse to the reveiewer specific comment on rural-urban variations in ANC use. We are considering an investigation into rural-urban differences in ANC use among Indian women.

Reference

Senanayake, P., O’Connor, E., & Ogbo, F. A. (2019). National and rural-urban prevalence and determinants of early initiation of breastfeeding in India. BMC public health, 19, 896.

Page 6: the text is easier to follow if the authors comment the data by the same order they are presented in Table 2. Table 2: I believe the odds presented concern “0 antenatal visits” as the reference category of the outcome variable. That should be explicitly stated.

Response:

Point appreciated and now reflected in the revised manuscript (Point 3.2 and Page 3, Paragraph 3):

No ANC visit group formed the reference category of the outcome variables in the analyses.

Also, I believe that it could be more informative to consider “4 or more antenatal visits” as the reference category. The pattern of results when comparing “0 antenatal visits” to “1-3 antenatal visits” and “4 or more antenatal visits” is largely similar. I would expect more differences if “4 or more antenatal visits” was compared with “0 antenatal visits” and with “1-3 antenatal visits”.

Response:

We believe that our study was informed by the focused research question, which is, what are the enablers and barriers to ANC service use among Indian women? If we are to use “4 or more antenatal visits” as the reference category, this would not answer our research question. We also note that similar research questions have been asked and answered in many published studies from LMICs:

De Allegri, M., Ridde, V., Louis, V. R., Sarker, M., Tiendrebéogo, J., Yé, M., ... & Jahn, A. (2011). Determinants of utilisation of maternal care services after the reduction of user fees: a case study from rural Burkina Faso. Health policy, 99(3), 210-218. Dairo, M. D., & Owoyokun, K. E. (2010). Factors affecting the utilization of antenatal care services in Ibadan, Nigeria. Benin Journal of Postgraduate Medicine, 12(1). Adamu, Y. M., & Salihu, H. M. (2002). Barriers to the use of antenatal and obstetric care services in rural Kano, Nigeria. Journal of obstetrics and gynaecology, 22(6), 600-603. Simkhada, B., Teijlingen, E. R. V., Porter, M., & Simkhada, P. (2008). Factors affecting the utilization of antenatal care in developing countries: systematic review of the literature. Journal of advanced nursing, 61(3), 244-260. Tsegay, Y., Gebrehiwot, T., Goicolea, I., Edin, K., Lemma, H., & San Sebastian, M. (2013). Determinants of antenatal and delivery care utilization in Tigray region, Ethiopia: a cross-sectional study. International journal for equity in health, 12(1), 30.

Pages 6-14: the authors should specify whether their focus is on crude odds ratios or adjusted odds ratios.

Response:

Point appreciated and now reflected in the revised manuscript (Page 6–15).

Discussion

Page 11: The term “internal validity” is not correctly used, as it is only present in experimental studies. The authors are referring to construct validity. 

Response:

We disagree with the reviewer that internal validity is only present in experimental studies. Our understanding is that “internal validity is the extent to which the results of a study reflect the true situation in the study sample in the absence of any alternative explanations”. These alternative explanations, namely chance, bias and confounding, have been the focus of this and the previous chapters. The prime objective of study design, implementation, analysis and interpretation is to maximise the internal validity of a study (Webb, Bain & Page, 2016). The study design is not specific to experimental studies as it could also be observational studies (e.g. case-control or cohort studies).

Reference

Webb, P., Bain, C., & Page, A. (2016). Essential epidemiology: an introduction for students and health professionals. Cambridge University Press.

Reviewer 2 Report

I maybe mistaken but please compare Lines 147-149 and 158-159

158-159 appears contradictory to line 147-149; Though there is a double negative used in that sentence.
148 those in the South reported the lowest proportion of non-use of ANC services.

I would suggest providing definitions for the various types of Castes/tribes you refer to: scheduled castes, scheduled tribes, other.. etc. For a reader unfamiliar with these terms, it would throw a great deal of light on these challenges the population has to face. 

Author Response

Comments and Suggestions for Authors

I maybe mistaken but please compare Lines 147-149 and 158-159

158-159 appears contradictory to line 147-149; Though there is a double negative used in that sentence. 
148 those in the South reported the lowest proportion of non-use of ANC services.

Response:

The reviewer has eyes for details – thank you for the essential observation! Line 147-149 relates to the proportion of no ANC visit and was incorrectly written but has now been corrected in the revised manuscript (Page 4, Paragraph 3).

Line 158–159 relates to the likelihood (odds) of women making between 1 and 3 ANC visits and was correctly written.

I would suggest providing definitions for the various types of Castes/tribes you refer to: scheduled castes, scheduled tribes, other.. etc. For a reader unfamiliar with these terms, it would throw a great deal of light on these challenges the population has to face.

Response

Point appreciated and now reflected in the revised manuscript (Page 3, Paragraph 4):

In India, the Constitution recognises certain ethnic minority groups for special consideration as Scheduled Tribes, Scheduled Castes or other backward classes [24, 25].

Reviewer 3 Report

This paper was extremely well done.  I did not find any need for corrections.  In the discussion or limitations section, I wonder if the authors could add something on the potential for using data from pregnant women and their current antenatal visit situation, which would depend, of course, on the length of their current pregnancy.  The sample size is so large, one might find a hundred or more such situations, which could be used to confirm/check against recall bias in the current study.

Author Response

Comments and Suggestions for Authors

This paper was extremely well done.  I did not find any need for corrections.  In the discussion or limitations section, I wonder if the authors could add something on the potential for using data from pregnant women and their current antenatal visit situation, which would depend, of course, on the length of their current pregnancy.  The sample size is so large, one might find a hundred or more such situations, which could be used to confirm/check against recall bias in the current study.

Response:

We thank the reviewer for the constructive comment. Great suggestion by the reviewer around the assessment of pregnant women and their current antenatal visit situation. We note that we are currently investigating the extent to which pregnant Indian women receive all recommended components of the ANC visit, which largely depends on their contexts. We agree with the reviewer that there might be a possibility of recall bias in the data but it is challenging to detect in secondary data like the NFHS-4. In the analysis, we made effort to reduce the potential effect of recall bias, consistent with past studies. This information was noted in the original manuscript (Page 3, Paragraph 2).